# Genetic Variants of the TERT Gene and Telomere Length in Obstructive Sleep Apnea

**DOI:** 10.3390/biomedicines10112755

**Published:** 2022-10-30

**Authors:** Piotr Macek, Rafal Poreba, Pawel Gac, Katarzyna Bogunia-Kubik, Marta Dratwa, Mieszko Wieckiewicz, Anna Wojakowska, Monika Michalek-Zrabkowska, Grzegorz Mazur, Helena Martynowicz

**Affiliations:** 1Department of Internal Medicine, Occupational Diseases, Hypertension and Clinical Oncology, Wroclaw Medical University, 213 Borowska St., 50-556 Wroclaw, Poland; 2Department of Population Health, Division of Environmental Health and Occupational Medicine, Wroclaw Medical University, Mikulicza-Radeckiego 7, 50-368 Wroclaw, Poland; 3Laboratory of Clinical Immunogenetics and Pharmacogenetics, Hirszfeld Institute of Immunology and Experimental Therapy, Polish Academy of Sciences, 50-422 Wroclaw, Poland; 4Department of Experimental Dentistry, Wroclaw Medical University, 26 Krakowska St., 50-425 Wroclaw, Poland

**Keywords:** TERT, single nucleotide polymorphism, telomerase, obstructive sleep apnea, diabetes, hypertension

## Abstract

Introduction: Obstructive sleep apnea (OSA) is a worldwide breathing disorder that has been diagnosed globally in almost 1 billion individuals aged 30–69 years. It is characterized by repeated upper airway collapses during sleep. Telomerase reverse transcriptase (TERT) is involved in the prevention of telomere shortening. This prospective, observational study aimed to investigate the relationship between single nucleotide polymorphisms (SNPs) of TERT and the severity of OSA, taking into account hypertension and diabetes prevalence. Methods: A total of 149 patients with OSA were diagnosed using one-night video-polysomnography based on the American Academy of Sleep Medicine guidelines. The TERT SNPs and telomere length (TL) were detected using real-time quantitative polymerase chain reaction. Results: Statistical analysis showed that there is no relationship between the rs2853669 and rs2736100 polymorphisms of TERT, and the severity of OSA (*p* > 0.05). Moreover, no relationship between TL and the severity of OSA was observed. The G allele in the locus of rs2736100 TERT was associated with hypertension prevalence and was more prevalent in hypertensives patients (46.00% vs. 24.49%, *p* = 0.011). The prevalence of hypertension was higher in patients with the C allele in the locus of rs2853669 than in patients without this allele (50.79% vs. 30.23%, *p* = 0.010). Moreover, a lower prevalence of diabetes was observed in homozygotes of rs2736100 TERT than in heterozygotes (5.63% vs. 15.38%, *p* = 0.039). Conclusion: This study showed no relationship between OSA and TERT SNPs. However, SNPs of the TERT gene (rs2736100 and rs2853669) were found to affect arterial hypertension and diabetes prevalence.

## 1. Introduction

Obstructive sleep apnea (OSA) is one of the most common respiratory system diseases. Its prevalence is continuously increasing, affecting nearly 1 billion adults aged 30–69 years worldwide [1]. OSA patients encounter repeated upper airway collapses, leading to oxygen desaturation, numerous arousals, and sleep fragmentation, which may influence the aging process [2]. The number of apnea and hypopnea episodes, arousals, and time during the night spent with an oxygen saturation level of less than 90% contribute to the alteration of cellular communication, deregulation of nutrient sensing, mitochondrial dysfunction, and genomic instability in OSA patients [3]. Snoring, apneas, and sleepiness are the main symptoms of OSA although fatigue, shortness of breath and choking, erectile dysfunction, trouble concentrating, and even insomnia are reported in some patients [4]. OSA leads to cardiovascular conditions such as arterial hypertension, stroke, coronary artery disease, heart failure, and arrhythmias [5]. About 40% of OSA patients also suffer from hypertension. However, about 50% of hypertensive patients [6] and 72–80% of patients with resistant hypertension meet the criteria for OSA (apnea–hypopnea index, AHI > 5) [7]. Endothelium dysfunction, systemic inflammation, and oxidative stress are common in both OSA and hypertension, promoting biological aging as well [8]. Diabetes also commonly co-occurs with OSA [9]. Intermittent hypoxia, which is a central feature of OSA, elicits a proinflammatory response in visceral adipose tissue and contributes to insulin resistance [10]. Therefore, OSA has been shown to cause or worsen metabolic syndrome [11].

Telomerase reverse transcriptase (TERT), encoded by the TERT gene, is a specialized reverse transcriptase composed of a catalytic protein (reverse transcriptase, TERT) and telomerase RNA components (TERC) [12]. It plays a vital role in maintaining telomere and chromosomal integrity and stability [13]. Several studies showed that single nucleotide functional polymorphisms (SNPs) in the TERT gene may affect its expression and telomerase activity [14]. SNPs at loci encoding TERT and TERC have been shown to be correlated with telomere length (TL) and an increased risk of age-related diseases and mortality [15,16,17,18]. TERT rs2736100 and TERC rs12696304 are two well-researched SNPs affecting telomerase expression and TL [19,20,21,22].

However, data on the relationship between SNPs of TERT and OSA are scarce in the literature. Therefore, this study aimed to evaluate the role of TERT SNPs (rs2736100 and rs2853669) in OSA, taking into account hypertension and diabetes prevalence. The null hypothesis was that the genetic variant of TERT varies between OSA patients and healthy controls, as well as between hypertensive and normotensive patients and between diabetic and nondiabetic patients.

## 2. Materials and Methods

### 2.1. The Study Design

This prospective, observational study was carried out on patients admitted to the Department and Clinic of Internal, Occupational Diseases, Hypertension, and Clinical Oncology of Wroclaw Medical University. One-night video-polysomnography was conducted in the Sleep Laboratory of Wroclaw Medical University using Nox-A1 (Nox Medical, Reykjavik, Iceland).

A total of 149 patients were enrolled in this study (44.67% male and 55.33% female). Average age and body mass index (BMI) was 49.00 ± 15.07 and 28.96 ± 5.05, respectively. Overweight (BMI > 25), diabetes mellitus, arterial hypertension, and ischemic heart disease were diagnosed in 46.30%, 11.40%, 39.59%, and 5.37% of individuals, respectively.

The group size was determined using a sample size calculator. The selection conditions were as follows: population size 3 million, fraction size 0.1, maximum error 5%, significance level 0.05. The required minimum size of the study group was 139. During the analyzed period, 149 patients at Sleep Lab were examined, hence the final size of the study group.

Patients who met the following inclusion criteria were included in this study: willingness to participate, clinically suspected OSA, and age > 18 years old. The exclusion criteria were as follows: severe mental disorders, which prevent them from undergoing polysomnography, intake of drugs that can affect the breathing and/or neuromuscular activity, active malignancy, respiratory and/or cardiac insufficiency, and active inflammation.

All participants provided informed consent to participate in the study, and the study was approved by the Ethics Committee of Wroclaw Medical University (ID KB 525/2020) and conducted following the Declaration of Helsinki.

Diagnosis of OSA was made based on the American Academy of Sleep Medicine (AASM) standards. All participants underwent one-night polysomnography. The following parameters were recorded: sleep and respiratory data such as sleep latency, wake after sleep onset, rapid eye movement (REM) latency, sleep efficiency, total sleep time, and the ratio of N1 (sleep stage 1), N2 (sleep stage 2), N3 (sleep stage 3), and REM (REM sleep stage), along with the video and audio recording of the sleep episode. Respiratory events were recorded using a nasal pressure transducer, and oxygen saturation was measured using a finger pulse oximeter. Following the one-night polysomnography, a certified polysomnographist assessed automatic 30-s epochs of polysomnograms, and the epochs were classified based on the standard criteria for sleep by the AASM 2013 Task Force. Respiratory events were documented as follows: the absence of airflow (>90%) for ≥10 s was scored as an apneic event, whereas a reduction in the amplitude of breathing by ≥30% for ≥10 s with a ≥3% decline in the blood oxygen saturation level or arousal was scored as a hypopnea. The respiratory effort was assessed using respiratory inductance plethysmography belts around the thorax and abdomen. A single modified electrocardiogram lead II was used to assess the ECG. On the day after polysomnography, blood samples of the patients were collected at 7.00 a.m. by venipuncture.

Arterial hypertension, ischemic heart disease, and diabetes mellitus were recorded based on medical history. All patients were weighed and measured on admission to the ward.

### 2.2. DNA Extraction

Genomic DNA was isolated from 200 µL of peripheral blood taken on EDTA using the NucleoSpin Blood (MACHEREY-NAGEL GmbH & Co. KG, Dueren, Germany) based on the manufacturer’s instructions. DNA concentration and purity were quantified using a DeNovix (DeNovix Inc., Wilmington, DE, USA). The isolated DNA was subjected to TERT genotyping and the evaluation of TL.

### 2.3. TL Quantification

The average TL was measured in the DNA samples of 149 patients. The DNA samples were diluted with nuclease-free water to a concentration of 5 ng/µL, and the TL was measured using a real-time quantitative polymerase chain reaction (qPCR) on a LightCycler480 II Real-Time PCR system (Roche Diagnostics International, Rotkreuz, Switzerland) using qPCR assay kits (ScienCell’s Absolute Human Telomere Length Quantification qPCR Assay Kit, Carlsbad, CA, USA), according to the manufacturer’s instructions. Two consecutive reactions were performed for each of the DNA samples: the first for amplification of the single-copy reference (SCR) gene and the second for the telomeric sequence. The SCR primer set is used to recognize and amplify the 100-bp region on human chromosome 17 and serves as a reference for data normalization. The qPCR conditions were as follows: 95 °C for 10 min, followed by 32 cycles of 95 °C for 20 s, 52 °C for 20 s, and 72 °C for 45 s. Data were analyzed according to the manufacturer’s instructions. All reactions were run in duplicate.

### 2.4. Genotyping of TERT Gene Polymorphisms

The SNPs within the TERT gene were chosen based on results from the SNP Function Prediction tool of the National Institute of Environmental Health Sciences (NCBI Database) website and other auxiliary databases (https://snpinfo.niehs.nih.gov/snpinfo/snpfunc.html; https://www.ncbi.nlm.nih.gov/snp/; https://www.ensembl.org/index.html). The following criteria were used: change in RNA and/or amino acid chain, potential splicing site and/or miRNA binding site, and minor allele frequency in Caucasians above 10%.

Based on the above criteria, two SNPs were selected for the study: TERT rs2736100 (G > T), located in intron 2, and TERT rs2853669 (T > C), located at −245 bp (Ets2 binding site) in the promoter region. These two SNPs were determined using LightSNiP typing assays (TIB MOLBIOL, Berlin, Germany). Both assays were based on qPCR. Amplifications were performed on a LightCycler480 II Real-Time PCR system (Roche Diagnostics International AG, Rotkreuz, Switzerland) according to the manufacturer’s instructions. The PCR conditions were as follows: 95 °C for 10 min, followed by 45 cycles of 95 °C for 10 s, 60 °C for 10 s, and 72 °C for 15 s. The PCR was followed by one cycle of 95 °C for 30 s, 40 °C for 2 min, and gradual melting from 75 °C to 40 °C.

The two TERT polymorphic variants were detected in patients and controls using LightSNiP typing assays (TIB MOLBIOL, Berlin, Germany) by real-time PCR amplifications with melting curve analysis. The reactions were performed on a LightCycler 480 II Real-Time PCR system (Roche Diagnostics International, Rotkreuz, Switzerland) following the manufacturer’s instructions.

### 2.5. Statistical Analysis

Statistical analysis was carried out using the statistical package “Dell Statistica 13.1” (Dell Inc., Round Rock, TX, USA). The arithmetic means and SDs of the estimated parameters were calculated for the quantitative variables. The distribution of the variables was evaluated using Lilliefors and Shapiro–Wilk W tests. For independent quantitative variables with normal and nonnormal distribution, Student’s t-test and Mann–Whitney U test were used, respectively. Based on these results, qualitative variables were expressed as percentages. The results at *p* < 0.05 were considered statistically significant.

## 3. Results

The participants were classified into two groups: the study group (patients with OSA) and the control group (patients without OSA). OSA was diagnosed in 100 patients (67.11%). Mild (5 ≤ AHI < 15), moderate (15 ≤ AHI < 30), and severe OSA (AHI ≥ 30) were diagnosed in 35 (23.48%), 29 (19.46%), and 36 (24.16%) patients, respectively. In 49 (32.89%) participants, OSA was excluded (AHI < 5). The average TL was 2.98 ± 1.07 kB. All parameters observed using polysomnography are presented in Table 1.

A negative linear correlation was observed between TL and age (r = −0.18, *p* = 0.028); however, no statistically significant correlation was found between TL and polysomnography parameters. Moreover, no statistically significant correlation or differences were found between TL and the diagnosis and severity of OSA.

Then, the relationship between the SNPs of TERT and OSA was evaluated. The prevalence of the TERT SNPs among the study participants is presented in Table 2.

No statistically significant differences between the TERT SNPs and more frequent OSA diagnoses were observed among the study participants. The prevalence of the TERT SNPs with the respect to the AHI value (patients with confirmed vs. excluded OSA) is presented in Table 3. Whether the TERT SNPs affect the severity of OSA was also evaluated. No relationship was observed between the nucleotide system of both polymorphisms and the incidence of mild, moderate, and severe OSA. The effect of TERT SNPs on TL was not found.

The relationship between arterial hypertension, diabetes mellitus and ischemic diseases, and TERT SNPs were studied. The information about these disorders was obtained based on medical history. In addition, the mean BMI values in patients with the studied polymorphisms were compared.

First, the patients were compared based on the rs2736100 SNP, in which significant differences between the TERT SNP and arterial hypertension diagnoses were observed. However, no statistical differences between polymorphisms and other diseases were found. All details are presented in Table 4.

Moreover, the results showed that patients with the G allele more commonly suffer from arterial hypertension than patients without the G allele (Table 5). Other alleles were not found to influence the prevalence of disorders. While analyzing the differences between patients based on homozygotes and heterozygotes of rs2736100, a homozygous nucleotide system was found to decrease the prevalence of arterial hypertension and diabetes mellitus. All details are presented in Table 6.

Then, the prevalence of the studied disorders was compared based on the rs2853669 SNP. Statistically significant differences were observed between the rs2853669 SNP and arterial hypertension alone. No relationship was found between other disorders and the rs2854669 SNP. All details are presented in Table 7.

Patients with the C allele in the rs2853669 SNP were more commonly diagnosed with arterial hypertension than those without it (Table 8). No relationships between other alleles, homozygotes, and the prevalence of the other studied disorders were observed.

Finally, no relationship was observed between the mean values of BMI and the TERT SNPs.

## 4. Discussion

Located on chromosome locus 5p15.33, TERT plays a role in the lengthening and preservation of telomeres. In the present study, two well-studied SNPs—rs2853669 in the TERT promoter and rs2736100 located within intron 2 of TERT—were investigated in a group of patients with high OSA risk. OSA is considered an aging disease, especially related to accelerated vascular aging. In the course of OSA, the accumulation of functional and structural changes in vessels, including arterial stiffening, increased carotid intima–media thickness, as well as carotid diameter enlargement, is an important contributor to cardiovascular diseases. The primary mechanism of accelerated vascular aging in OSA is complex and includes intermittent hypoxia, sympathetic activation, and systemic inflammation [23]. Yagihara et al. classified participants in their study into a group of patients treated with CPAP (continuous positive airway pressure) and a control group treated with a nasal dilator and reported that the age of patients with OSA judged by appearance was lower in the group treated with CPAP compared with the control group [24].

Previous studies reported that the TERT variant rs2736100 is associated with TL and multiple disease risks, including colorectal cancer, myeloproliferative neoplasms [21], primary glomerulonephritis/end-stage renal disease [25], decreased idiopathic pulmonary fibrosis, combined pulmonary fibrosis, and emphysema syndrome [26]. The presence of the C allele in the locus of the rs2736100 SNP of the TERT gene is associated with longer telomeres, whereas the presence of the A allele is associated with shorter telomeres. The meta-analysis showed that cancer was positively associated with the C allele and that noncancerous diseases were negatively associated with the C allele [27].

To the best of our knowledge, this is the first study investigating the role of rs2736100 and rs2853669 genetic variants in OSA. Although no relationships were observed between OSA and the TERT SNPs, an association between hypertension and TERT gene polymorphism was observed, as well as an association between diabetes and TERT gene polymorphism. Hypertension is a common age-related disease that increases the risk of cardiovascular diseases, including atherosclerosis, coronary heart disease, stroke, and chronic kidney disease [8]. In patients with essential hypertension, a shorter leukocyte TL coupled with a lower expression of telomerase genes was reported previously. However, no differences in the genotype of rs2736100 and rs12696304 SNPs were reported between normotensive and hypertensive patients [28]. Feng showed that homozygous GG was associated with atherosclerosis risk [29]. Hypertension and diabetes are risk factors for atherosclerosis development. Thus, the findings of the present study are in agreement with these observations. The results of the present study showed that the G allele of the rs2736100 SNP was associated with hypertension and diabetes prevalence and that it was more prevalent in hypertensive patients; however, no statistical differences were observed in the T allele. In addition, differences were found in the rs2853669 SNP, taking into account the presence of the C allele in its locus. (50.79% vs. 30.23%, *p* = 0.010).

OSA is commonly associated with diabetes, with prevalence ranging from 23% to 86% [30]. However, comorbid OSA is observed between 30% [31] and 80% [32] of people with diabetes. Other common comorbidities of diabetes include heart disease, stroke, cancers, dementia, kidney disease, and depression [33]. The aging process in diabetes is associated with alterations in glucose metabolism, including both relative insulin resistance and islet cell dysfunction, thus leading to impaired glucose intolerance and/or postprandial hyperglycemia [34]. A previous study demonstrated that the incidence of type 2 diabetes was increased in carriers of the CC SNP (rs2853669) of the TERT gene [35]. Goswami showed that the TC genotype plays a protective role against the development of type 2 diabetes [36]. The present study showed a decreased prevalence of diabetes in homozygotes of the rs2736100 SNP of the TERT gene (5.63% vs. 94.37%, *p* = 0.039). Thus, the results of the present study are in agreement with those of previous studies.

TL was also evaluated in a patient with OSA. Telomeres are DNA regions of variable length at the end of chromosomes that protect the chromosome and prevent DNA from degradation [37,38]. A part of telomeric DNA is lost during cell division. Therefore, TL is an indicator of biological aging [38,39]. Goglin et al. [40] reported the correlation between mortality rate and TL. Telomeres are often referred to as a “molecular clock of aging” because of their progressive shortening due to aging [41]. TL may be affected by various factors, such as gender [42], psychological stress [43], nutritional factors [44,45], physical activity, and TERT activity. Individuals with a short TL were at an increased risk of cardiovascular diseases, myocardial infarction, heart failure, and stroke [46,47,48,49,50]. Intermittent hypoxia also directly affects TL [51]. The activity of HIF-1 (hypoxia-inducible factor, a heterodimer composed of subunits α and β) is found to be increased in OSA patients [52,53,54,55]. The overactivation of HIF-1 leads to overexpression of TERT, an increased level of telomerase, and telomere stabilization [56,57,58]. The data on the association between TL and OSA are contradictory. Many studies have reported LTL shortening in OSA [59,60,61,62,63]; however, a few studies have reported telomere lengthening [64,65,66].

In this study, no significant relationship between the TERT gene polymorphisms and average BMI values was observed. Shen et al. [67] reported no effect modifications between the variant alleles and breast cancer risk in subgroups stratified by cigarette smoking, BMI status, and family history.

No correlation between TL and OSA, hypertension, diabetes, or ischemic heart disease was observed in the present study; however, a positive linear correlation between TL and age was found. These observations may be attributable to the bidirectional effect of OSA on telomeres depending on the disease’s onset, severity, and duration. Initially, an increase in the HIF-1 telomerase activity protects telomeres, but when the oxidative stress and the inflammatory state reach the threshold level, telomere shortening begins. However, this hypothesis needs further research. The following are the limitations of the present study: the subgroup of diabetes was small, and ambulatory blood pressure monitoring was not conducted; however, this is beyond the scope of the study.

## 5. Conclusions

No association between OSA and TERT SNPs was observed.The SNPs of the TERT gene (rs2736100 and rs2853669) were found to affect arterial hypertension prevalence.The prevalence of type 2 diabetes was decreased in homozygotes of the rs2736100 SNP of the TERT gene.

## Figures and Tables

**Table 1 biomedicines-10-02755-t001:** Polysomnographic data of the study sample and telomere length.

Parameter	Mean	Minimum	Maximum	SD
AHI (n/h)	18.50	0.2	100	20.54
ODI (n/h)	17.56	0.1	88.40	19.28
SL (min)	21.86	1	112.60	20.23
WASO (min)	52.29	1.00	195.50	44.75
SE (%)	82.69	52.40	98.30	10.28
NREM 1 (%of TST)	4.75	0.1	36.10	4.79
NREM 2 (% of TST)	46.73	4.30	78.70	11.43
NREM 3 (% of TST)	26.00	2.60	54.90	10.82
REM (% of TST)	22.23	4.10	41.30	7.57
MEAN SpO2 (%)	93.08	83.30	97.30	2.43

SD—standard deviation; AHI—apnea–hypopnea index; ODI—oxygen desaturation index; SL—sleep latency; WASO—wake after sleep onset; SE—sleep efficiency; NREM– sleep stage 1; NREM2—sleep stage 2; NREM3—sleep stage 3; REM—rapid eye movement sleep stage; MEAN SpO2—mean oxygen saturation.

**Table 2 biomedicines-10-02755-t002:** The prevalence of TERT SNP in the entire study group.

**TERT rs2736100**			**TERT rs2853669**		
Genotype	*n*	%	Genotype	*n*	%
TG	78	52.34	TT	86	57.72
TT	49	32.89	TC	52	34.90
GG	22	14.77	CC	11	7.38
Allele			Allele		
T	127	85.23	C	63	42.28
G	100	67.11	T	138	92.62

**Table 3 biomedicines-10-02755-t003:** The prevalence of the TERT SNP’s in patients with and without OSA. Statistically significant differences are indicated by boldface (*p* < 0.05).

TERT rs2736100				TERT rs2853669			
Genotype	AHI ≥ 5	AHI < 5	Average telomere length [kB]	Genotype	AHI ≥ 5	AHI < 5	Average telomere length [kB]
TG	55 (70.51%)	23 (29.49%)	2.97 ± 1.03	TT	58 (68.24%)	27 (31.76%)	3.01 ± 1.10
TT	28 (58.33%)	20 (41.67%)	2.93 ± 1.14	TC	36 (69.23%)	16 (30.77%)	2.97 ± 1.10
GG	16 (72.73%)	6 (27.27%)	3.08 ± 1.09	CC	5 (45.45%)	6 (54.55%)	2.70 ± 0.68

**Table 4 biomedicines-10-02755-t004:** The prevalence of arterial hypertension in the rs2736100 SNPs of the TERT gene. Statistically significant differences are indicated by boldface (*p* < 0.05).

Allele	Hypertensives	Normotensives	Diabetics	Non-Diabetics	BMI Value	Ischemic Disease	Non-Ischemic Disease
TG	**37 (47.44%)**	**41 (52.56%)**	12 (15.38%)	66 (84.62%)	28.69 ± 4.36	6 (7.69%)	72 (92.31%)
TT	**12 (24.49%)**	**37 (75.51%)**	3 (6.12%)	46 (93.88%)	30.47 ± 6.99	2 (4.08%)	47 (95.92%)
GG	**9 (40.91%)**	**13 (59.09%)**	1 (4.55%)	21 (95.45%)	27.38 ± 3.45	0 (0.00%)	22 (100%)

**Table 5 biomedicines-10-02755-t005:** The prevalence of arterial hypertension among patients carrying the G allele in the locus of rs2736100.

	Hypertensives	Normotensives	*p* Value
G allele −	12 (24.49%)	37 (75.51%)	*p* = 0.011
G allele +	46 (46.00%)	54 (54.00%)	

**Table 6 biomedicines-10-02755-t006:** The prevalence of the disorders in patients with the rs2736100 SNP of the TERT gene, taking into account the presence of homozygotes. Statistically significant differences are indicated by boldface (*p* < 0.05).

	Hypertensives	Normotensives	Diabetics	Non-Diabetics	BMI Value	Ischemic Disease	Non-Ischemic Disease
Homozygous (+)	**21 (29.58%)**	**50 (70.42%)**	**4 (5.63%)**	**67 (94.37%)**	29.29 ± 6.02	2 (2.82%)	69 (97.18%)
Homozygous (−)	**37 (47.44%)**	**41 (52.56%)**	**12 (15.38%)**	**66 (84.62%)**	28.69 ± 4.37	6 (7.69%)	72 (92.31%)

**Table 7 biomedicines-10-02755-t007:** The prevalence of arterial hypertension in the rs2853669 SNPs of the TERT gene. Statistically significant differences are indicated by boldface (*p* < 0.05).

Allele	Hypertensives	Normotensives	Diabetics	Non-Diabetics	BMI Value	Ischemic Disease	Non-Ischemic Disease
TT	**26 (30.23%)**	**60 (69.77%)**	8 (9.30%)	78 (90.70%)	29.40 ± 5.87	3 (3.49%)	83 (96.51%)
TC	**27 (51.92%)**	**25 (48.08%)**	8 (15.38%)	44 (84.62%)	28.47 ± 4.11	5 (9.62%)	47 (90.38%)
CC	**5 (45.45%)**	**6 (54.55%)**	0 (0.00%)	11 (100%)	27.80 ± 2.68	0 (0.00%)	11 (100%)

**Table 8 biomedicines-10-02755-t008:** The prevalence of arterial hypertension among patients with the C allele in the locus of rs 2853669.

	Hypertensives	Normotensives	*p* Value
C allele −	26 (30.23%)	60 (69.77%)	*p* = 0.010
C allele +	32 (50.79%)	31 (49.21%)	

## Data Availability

The data presented in this study are available on request from the corresponding author.

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
