# Peer review of "Genetic Variants of the TERT Gene and Telomere Length in Obstructive Sleep Apnea"

_biomedicines, 2022, doi:10.3390/biomedicines10112755_

Round 1

Reviewer 1 Report

The study very thoroughly performed, with well described M&M, clear presentation of the results, detailed discussion of the relevant literature and clear conclusions. I have no criticisms to object.

Author Response

Dear Reviewer,
On behalf of our research team, I would like to attach  thank you for your time and thorough review of our manuscript. 

Reviewer 2 Report

This study by Piotr Macek and colleagues evaluated the role of TERT SNPs and obstructive sleep apnea (OSA) in a total of 149 individuals. No relationship between telomere length or SNPs of TERT and severity of OSA was identified.

Here are my comments:

1.      My biggest concern is the sample size of this study. The study only included a total of 149 individuals; however, I don’t think it is large enough to get a solid conclusion as the allele frequencies of the two SNPs (rs2736100 and rs2853669) didn’t consistent with those reported in the large databases. According to Table 2, the minor allele frequency (MAF) of rs2736100 is G=0.409, however, this value is C=0.499 (N=1006) in 1000Genomes Europe or and C=0.510 (N=237028) in Allele Frequency Aggregator Europe. It’s better to add a sample size calculation section. Moreover, the alleles should better be reported in the forward orientation.

2.      Many numbers in the tables are very confusing. How did the authors calculate the n of each allele? According to Table 2, the number of the G allele of rs2736100 should be 22*2+78=122. I think all the numbers and percentages of alleles are not correct. Moreover, the number of TT is 48 in table 3 and 49 in table 2.

3.      The authors should provide accurate P-values whenever possible. And multiple testing corrections should also be considered.

4.      The authors would benefit from a proofreading service to ensure that the language is correct and precise. For example, the word ‘a’ in the abstract (line 23) should be deleted.

Author Response

Dear Reviewer,

On behalf of our research team, I would like to attach our detailed response below. I would like to thank you for your review of our article and your suggestions which undoubtedly helped us to improve our manuscript.

Comments and Suggestions for Authors:

This study by Piotr Macek and colleagues evaluated the role of TERT SNPs and obstructive sleep apnea (OSA) in a total of 149 individuals. No relationship between telomere length or SNPs of TERT and severity of OSA was identified.

  1. My biggest concern is the sample size of this study. The study only included a total of 149 individuals; however, I don’t think it is large enough to get a solid conclusion as the allele frequencies of the two SNPs (rs2736100 and rs2853669) didn’t consistent with those reported in the large databases. According to Table 2, the minor allele frequency (MAF) of rs2736100 is G=0.409, however, this value is C=0.499 (N=1006) in 1000Genomes Europe or and C=0.510 (N=237028) in Allele Frequency Aggregator Europe. It’s better to add a sample size calculation section. Moreover, the alleles should better be reported in the forward orientation.

Response: According to your suggestions we have added a section on sample size calculation.

  1. Many numbers in the tables are very confusing. How did the authors calculate the n of each allele? According to Table 2, the number of the G allele of rs2736100 should be 22*2+78=122. I think all the numbers and percentages of alleles are not correct. Moreover, the number of TT is 48 in table 3 and 49 in table 2.

Response: In the first stage, we counted patients with a specific genotype, and in the next stage, we counted the number of patients with a specific allele in the locus. The difference in the number of patients in Table 2 and Table 3 is due to the lack of AHI values for technical reasons for one patient qualified for the study group.

  1. The authors should provide accurate P-values whenever possible. And multiple testing corrections should also be considered.

Response: we have provided p-values whenever possible thank you for your notice.

  1. The authors would benefit from a proofreading service to ensure that the language is correct and precise. For example, the word ‘a’ in the abstract (line 23) should be deleted.

Response: Thank you for your notice, the text of the manuscript was checked by a native speaker.

Round 2

Reviewer 2 Report

I don't have further suggestions.